# Molecular Mechanisms of Microglial Motility: Changes in Ageing and Alzheimer’s Disease

**DOI:** 10.3390/cells8060639

**Published:** 2019-06-25

**Authors:** Diana K. Franco-Bocanegra, Ciaran McAuley, James A. R. Nicoll, Delphine Boche

**Affiliations:** 1Clinical Neurosciences, Clinical and Experimental Sciences, Faculty of Medicine, University of Southampton, Southampton SO16 6YD, UK; D.K.Franco-Bocanegra@soton.ac.uk (D.K.F.-B.); Ciaran.Mcauley@uhs.nhs.uk (C.M.); j.nicoll@soton.ac.uk (J.A.R.N.); 2Department of Cellular Pathology, University Hospital Southampton NHS Foundation Trust, Southampton SO16 6YD, UK

**Keywords:** Microglia, motility, morphology, ageing, Alzheimer’s disease

## Abstract

Microglia are the tissue-resident immune cells of the central nervous system, where they constitute the first line of defense against any pathogens or injury. Microglia are highly motile cells and in order to carry out their function, they constantly undergo changes in their morphology to adapt to their environment. The microglial motility and morphological versatility are the result of a complex molecular machinery, mainly composed of mechanisms of organization of the actin cytoskeleton, coupled with a “sensory” system of membrane receptors that allow the cells to perceive changes in their microenvironment and modulate their responses. Evidence points to microglia as accountable for some of the changes observed in the brain during ageing, and microglia have a role in the development of neurodegenerative diseases, such as Alzheimer’s disease. The present review describes in detail the main mechanisms driving microglial motility in physiological conditions, namely, the cytoskeletal actin dynamics, with emphasis in proteins highly expressed in microglia, and the role of chemotactic membrane proteins, such as the fractalkine and purinergic receptors. The review further delves into the changes occurring to the involved proteins and pathways specifically during ageing and in Alzheimer’s disease, analyzing how these changes might participate in the development of this disease.

## 1. Introduction

Microglia are the primary tissue-resident immune cells of the central nervous system, where they comprise between 0.5% to 16% of the total number of cells in the human brain, depending on the anatomical region [1]. During embryogenesis, microglia derive from the yolk sac [2] and are identified as a distinct cell population, displaying a unique gene expression profile [3].

Throughout the lifespan, microglia carry out a broad diversity of functions. During brain development, they contribute to shape the synaptic architecture, and they are essential for the establishment and maturation of neural networks [4]. In the adult brain, microglia perform a constant surveillance of the brain parenchyma [5], where they constitute the first line of defense against pathogens or injury.

Appreciation of the role of microglia as part of the immune system is in fact a relatively recent discovery, considering that they were first described over 100 years ago by Franz Nissl [6]. They were not thought to carry out immune functions until several decades later when it was observed that they can express the cytokine interleukin (IL)-1 [7]. It is acknowledged that microglia have in common with peripheral macrophages the expression of several proteins including CD11b [8], CD14 [9], colony-stimulating factor 1 receptor (CSF1R) [10], CD68, human leukocyte antigen (HLA)-DR, macrophage scavenger receptor (MSR)-A, and the Fcγ receptors [11]. Like the macrophages, microglia are able to express a wide variety of proteins specialized in the recognition of “alarm signals”, that initiate mechanisms of cell defense, e.g., toll-like receptors (TLRs), nucleotide-binding oligomerization domain (NOD) proteins, and C-type lectin receptors [12,13].

In the adult brain, besides their role in immune response, evidence from studies in mice suggests that microglia have a role in synaptic plasticity, pruning dendritic spines, and presynaptic axon terminals. By these means, microglia seem to contribute to experience-dependent modification or elimination of synapses in sensory circuits [14], and/or to reshape synaptic connections after pathological events [15]. However, the latter remains controversial [16] and has not been observed in humans.

The morphology of microglia can be regarded as a spectrum from highly ramified to amoeboid cells. A ramified shape is defined by a small, rather round cell body with several long branching processes extending within the local environment. At the other extreme, amoeboid microglia can have an enlarged, amoeboid-like cell body with abundant cytoplasm and no processes, morphologically resembling a peripheral macrophage. In humans, microglia display a wide range of shapes along this spectrum [17] (Figure 1).

For a long time, it was thought that in the absence of inflammation or injury, microglia were in an inactive, resting state. However, direct observation of microglial movement in vivo using two-photon microscopy in mice, has revealed that microglia show a constant *baseline motility* [5]. This motility, also referred to as “surveillance” motility, is defined by the extension, retraction, and movement of the microglial processes, without migration of the cell body. This kind of movement allows microglia to survey their environment, to clear cellular debris, to interact with neurons and other glial components, and to remodel the extracellular matrix [18]. Studies have shown that surveillance motility is highly dependent of morphological factors such as the number, the length, and the degree of ramification of the microglial processes [19].

If, while performing its surveillance activity, a microglial cell encounters any substance that could be indicative of infection, insult or tissue damage, the cell will then exhibit what is called *directed motility*, also known as “chemotactic motility” [20]. This form of motility consists of a targeted extension of processes toward the source of injury. Interestingly, when microglia perform directed motility, the processes on the other side of the *soma* from the damaged area stop their surveillance activity and retract, indicating that these two forms of motility are mutually exclusive in a single microglial cell, i.e., they cannot take place at the same time [19]. While microglial motility most commonly pertains to the movement of the processes, there is also evidence of movement of the *soma* during directed motility, although it occurs in a much slower rate. It was observed that *soma* movement increases with age, and may be associated with a more proinflammatory phenotype [21].

Both baseline and directed motility result from a complex molecular machinery that mainly comprises mechanisms of reorganization of the actin cytoskeleton, geared together with a “sensory” system of membrane receptors that allow the cells to perceive changes in their microenvironment and respond in accordance [19].

## 2. Dynamics of the Microglial Actin Cytoskeleton

The cytoskeleton is a structure present in every cell, including the prokaryotic cells of bacteria and archaea [22]. It is involved in many aspects of cellular life, such as differentiation, replication, signaling, maintenance of the cell shape, cell motility, and apoptosis [23]. This well conserved structure is composed of three main structural elements: (i) *Microfilaments*, which in all eukaryotic cells are formed of actin; (ii) *microtubules*, formed of tubulin; and (iii) *intermediate filaments*, which are formed by a variety of proteins depending on the cell type [24]. Based on the length of these filaments and the complexity of their interconnections, the cell can control its mechanical properties, changing the cytoplasm from a viscous fluid to an elastic gel, and vice-versa [25].

As highly motile cells, microglia have a remarkable ability to change their shape in order to carry out all of their functions. This allows them to adapt to their environment, moving rapidly through narrow spaces within the parenchyma, and phagocytosing a variety of particles. These shape changes rely on the rearrangement of the cytoskeletal proteins, in particular the actin microfilaments.

In the microglial cell, one layer of actin, known as the *cell cortex*, fully covers the inner face of the cytoplasmic membrane, whereas the cytoplasm itself is populated by a three-dimensional network of actin filaments. To control motility and migration, the cell reorganizes these filaments to form different structures known as the *lamellipodia*, the *filopodia* and the *uropods* (Figure 2A). “Lamellipodia” are membrane-enclosed, very thin sheets of cytoplasm containing densely packed actin filament networks that arrange beneath the leading edge membrane. Continuous lamellipodium protrusion and ruffling is frequently accompanied by the formation of bundles of parallel actin filaments, most frequently termed “filopodia”. Filopodia play important roles in intercellular signaling, guidance toward chemoattractants and adhesion to the extracellular matrix. These two structures control the progression of the cell toward the origin of the chemoattractant signal; whereas another structure, named “uropod”, drags the rest of the cell body. The uropod is the rearward part of the cell, formed of a distinctive contractile trailing protrusion [26,27,28,29].

The actin filament (F-actin) is the basic building block of the actin cytoskeleton architecture, formed by the polymerization of globular actin monomers (G-actin) (Figure 2B). The microglial mechanisms of F-actin assembly, branching, cross-linking with other proteins and disassembly are described below. To form F-actin, the G-actin monomers first gather into oligomers, a step known as nucleation. After nucleation, oligomers further polymerize in a process dependent on the concentration of available G-actin in the cytoplasm. Actin polymers assemble together forming complex branched networks. The formation of branches in the actin network is an important process. Actin filaments need to be branched in order to form the lamellipodia and thus branching is essential for directed motility [30]. The process of branching is mainly controlled by an assembly of seven proteins, together known as the Arp2/3 complex (Figure 3A). Highly conserved among eukaryotes, the Arp2/3 complex consists of actin-related proteins (Arps) 2 and 3, and five additional subunits referred to as p41-Arc, p34-Arc, p21-Arc, p20-Arc, and p16-Arc, according to their molecular weights [31]. The Arp2/3 complex localizes to lamellipodia and filopodia, at the leading edge of moving cells [32]. In the presence of adenosine triphosphate (ATP), the complex binds to the side of a filament and initiates a “subfilament” which stems from the “mother filament” at a characteristic angle of 70° [33,34]. The formation of branches by the Arp2/3 complex is regulated by several mechanisms, which vary among cell types. In microglia, the proteins known as coronins seem to have an important role in Arp2/3 regulation. Coronins are part of the WD40 family of proteins. The WD40 repeat domain characteristically ends with a tryptophan-aspartate dipeptide to serve as a platform of protein–protein or protein–DNA interaction [35,36]. The mammalian genome contains seven coronin family members. The expression of one of them, coronin-1a (CORO1A), is highly abundant in microglia, has high affinity to F-actin [37], and has been observed to label microglia in animal and human brain (Figure 4A) [38].

CORO1A physically associates with Arp2/3 which is recruited to the ends of actin filaments [37] (Figure 3B). Evidence from a yeast model shows CORO1A is able to both activate and inhibit Arp2/3 branching activity, depending on its concentration. At low concentration, side-binding sites-adjacent to the bound CORO1A are free, and the Arp2/3 complex can be recruited to these sites; whereas at high concentration, the complex binding sites are blocked, and branching is inhibited as a result [39].

Other proteins important in actin assembly, in a non-microglial specific form, are the integrins, a family of adhesion receptors [40] which interact with actin through other protein mediators such as talin 1 (TLN1), α-actinin (ACTN1), and vinculin (VCL) [41,42,43].

In order to control the cell shape and movement, actin polymers interact with other proteins in a mechanism known as cross-linking (Figure 3C). A cross-linked actin network is defined as a structure formed of actin filaments connected with each other through proteins that bridge them together. Cross-linking helps to give adaptability to the actin network allowing the network to shape into more complex structures [26].

Several cross-linking proteins have been reported, differently expressed depending on the cell type. In microglia, the most notable is the ionized calcium-binding adapter molecule (Iba1), also known as allograft inflammatory factor 1 (AIF1), a protein involved in actin bundling and membrane ruffling [44]. Actin bundling is a cross-linking process in which the actin filaments are aligned in a parallel arrangement, packed tightly together. Actin bundles are thought to function as scaffolds to support lamellipodia, filopodia, and membrane ruffles, structures essential for microglial migration and phagocytosis [45,46]. Iba1 expression has been found exclusively restricted to monocytes, macrophages, and microglia. In these cells, expression of Iba1 is abundant and widespread along most of the cell cortex, including the highly ramified microglial processes present in their “baseline” state. Because of its presence in most, if not all microglial cells, and its broad distribution along the cell body and processes, Iba1 has become the most frequently used microglial marker (Figure 1 and Figure 4B) [47].

Another important microglial cross-linking protein is the non-muscle myosin II (NM II). This protein is essential for the contractile properties of the cytoskeleton [48]. The NM II protein consists of two heavy chains, connected by α-helical coiled-coil regions at the C-terminus, and two pairs of light chains, designated as “regulatory light chain” and “essential light chain”. The activity of this protein is regulated by phosphorylation by the myosin light chain kinase (MLCK) or Rho kinases (ROCKs). There are three NM II heavy chain isoforms in mammals, which determine the NM II isoforms (NM IIA, NM IIB, and NM IIC). The isoform NM IIB is the most widely expressed in the brain. In a mouse study, it was shown that treatment with the NM IIB inhibitor blebbistatin led to motility deficits in primary cultured microglia, as observed by reduced migration and phagocytosis [49].

Other cross-linking proteins expressed by microglia, but not exclusive to this cell type, are filamin C (FLNC) and spectrin which serve as a bridge to connect the actin cytoskeleton to the cytoplasmic membrane glycoproteins [50].

In order for the cell to perform its functions and respond to changes in the environment, the actin network is constantly reshaping itself. To achieve this, mechanisms of disassembly as well as assembly are required. The protein known as cofilin 1 (CFL1) is an important part of the disassembly machinery in all cells (Figure 3D), and in the human brain it is highly expressed in microglia (Figure 4C). CFL1 is an actin-binding protein that depolymerizes and severs actin filaments, in this way generating free G-actin monomers which are further recruited for filament elongation and branching. This function makes CFL1 one of the most important regulators of actin dynamics. CFL1 is activated and de-activated through a molecular “switch” modulated by the slingshot phosphatase (SSH1) and the LIM kinases. SSH1 activates the protein by dephosphorylation at serine 3, while LIM kinases have the opposite effect by deactivating it through phosphorylation [51,52]. Phosphorylation of CFL1 by LIM kinase participates in Rho-induced reorganization of the actin cytoskeleton [53]. In vitro studies have shown that CFL1 knockdown inhibits LPS-induced microglial activation and cell migration [54], highlighting the importance of this protein in microglial function.

## 3. Sensing the Environment: Role of the Chemotactic Membrane Receptors

As described above, the cytoskeleton provides the cell with the structures required for movement. However, how does the cell “know” the direction in which to conduct its movement? Microglia are equipped with a sophisticated system of membrane receptors, which provide a means for the cell to “sense” its environment, and therefore be able to guide their actin assembly toward the appropriate direction. This system of receptors has been referred to as the microglial “sensome” [55]. Evidence from mouse and rat studies has provided information about several membrane receptors that play an important role in the microglial ability to perceive microenvironmental changes and to trigger a motile response.

One of these is the fractalkine receptor (CX3CR1), which in physiological conditions is expressed exclusively by microglia [56]. Fractalkine, also known as CX3CL1, is a chemokine secreted by neurons. It is the only member of the ∂ (CX3C) chemokine family characterized by the presence of 3 amino acid residues (X3) located between 2 cysteine residues, forming a disulphide bond, a CX3C motif. Fractalkine exists in two forms: Membrane-bound and soluble [57]. Membrane-bound fractalkine acts as an adhesion protein, whereas the diffusible form is a chemoattractant protein [58]. Microglial baseline and directed motility were found to be significantly reduced in CX3CR1-GFP homozygous mice (CX3CR1 knockout (KO)/KO), compared to heterozygous controls (CX3CR1KO/+) [59].

One of the physiological functions of microglia is to clear cellular debris released from neurons and glial cells. This debris contains nucleosides such as ATP, adenosine diphosphate (ADP), uridine triphosphate (UTP) and uridine diphosphate (UDP), which when released to the extracellular matrix, have an important role in guiding microglial motility through the activation of microglial purinergic receptors. These receptors can be divided into two families based on their stimulus, activated by adenosine or ATP/ADP nucleotides, respectively the P1 (adenosine receptors) and P2 receptors (Figure 5).

Adenosine receptors are metabotropic and G-protein coupled, and four types have been characterized: A1, A2A, A2B, and A3 [60]. They are all expressed by microglia [61,62,63,64]. When ATP is released to the extracellular space as a result of cell death, it is rapidly sequentially hydrolyzed to ADP, AMP and adenosine by the action of the enzyme ectonucleoside pyrophosphatase-1 (ENPP1), and this in turn causes the activation of purinergic receptors P1 [65], a mechanism also known to trigger microglial directed motility [66]. In addition, cell culture studies have identified a role for the receptors A1, A2A, and A3 in microglial migration, with A2A involved in process retraction and A3 in process extension [62,67,68].

The P2 receptors are sensitive to the nucleosides triphosphate and diphosphate, and are divided in two families: The ionotropic P2X with seven members identified, and the metabotropic P2Y receptors with eight members identified [69,70]. From these, microglial cells express the receptors P2X4, P2X7, P2Y1, P2Y2, P2Y4, P2Y6, P2Y12, and P2Y13 [71,72,73].

P2X4 is an ATP/ADP receptor and in vitro studies have shown its involvement in microglial motility. Pharmacological blockade of this receptor by antagonists or by P2X4 short hairpin RNA interference inhibited chemotaxis in cultured rat microglia [74]. This study also showed that P2X4 regulation of microglial motility appears to be through a mechanism that involves the phosphatidylinositol-3-kinase (PI3K)/Akt pathway. Activation of P2X4 causes an increase in intracellular calcium, followed by activation of an Src kinase which recruits and activates PI3K. This in turn increases the concentrations of phosphatidylinositol-(3,4,5)-trisphosphate (PIP3), which results in the recruitment, phosphorylation, and activation of the regulatory kinase Akt [75]. This kinase is capable of inducing cytoskeletal rearrangements, as observed during cancer cell migration [76], but the mechanism in microglia remains unclear.

Regarding the P2X7 receptor, an ATP receptor, its activation results in the opening of a transmembrane channel that is permeable to Na^+^, K^+,^ and Ca^2+^ cations. Its main function in the cells is related to IL1 secretion. An efflux of K^+^ promotes activation of the IL1-converting enzyme, which cleaves pro-IL1 into its mature form, after which it is released from the cell [77]. The link between P2X7 and microglial motility is not clear, although it has been postulated that it may operate through calcium uptake which in turn activates the PI3K pathway, in a similar way to P2X4. It is also known that P2X7 interacts with Rho effector kinases, which are known to induce cytoskeletal arrangements [78]; however, these pathways have not been studied in microglia. This receptor is also involved in the α-secretase-dependent processing of the amyloid precursor protein (APP) [79], with implications for Alzheimer’s disease (AD), which are discussed below in later sections.

The P2Y receptors are G protein-coupled receptors which modulate the activity of associated voltage-gated K^+^ or Ca^2+^ channels [80].

P2Y1 is an ADP receptor associated with microglial cell migration [81]. P2Y1 is coupled to G-proteins of the G_q_ or G_11_ type. From in vitro studies in other cell types, it is known that the activation of these types of G-protein increases the levels of phospholipase C and inositol triphosphate, and down-regulates adenylate cyclase 5 (AC5) activity [82]. This downregulation promotes actin remodeling through up-regulation of CFL1 via adenylate cyclase-associated protein 1 (CAP1) [83].

P2Y2 is a UTP receptor, which modulates a mechanism of ion exchange between the cell and the extracellular matrix. A study using primary cultured rat microglia showed that P2Y2 binding to UTP promotes the G-coupled release of Ca^2+^ into the cytoplasm, from intracellular stores in the mitochondria and endoplasmic reticulum, causing a rise in cytoplasmic Ca^2+^ concentration. This stimulates Ca^2+^ release-activated Ca^2+^ (CRAC) channels, which allow an inward Ca^2+^ current to replenish the intracellular stores. This in turn induces calcium-activated potassium channels KCa3.1 to activate and release K^+^ out of the cell. UTP-stimulated microglial migration was inhibited by blocking either KCa3.1 or CRAC channels, highlighting the involvement of this mechanism in microglial motility [84].

P2Y4 is an ATP receptor involved in pinocytosis, a form of endocytosis that consists of the vesicle-mediated cellular “ingestion” of liquid from the extracellular space. In physiological conditions, a participation of this receptor in microglial motility or chemotaxis has not been established. Nevertheless, its role in pinocytosis requires a downstream pathway that results in actin reorganization for vesicle formation. Evidence from mouse studies shows that this downstream signaling can be through the PI3K/Akt pathway or via Rho signaling. The endocytic properties of this receptor have important implications for microglial Aβ phagocytosis, discussed below in the context of AD [56].

Another P2Y receptor associated with motility described in the mouse brain is P2RY6, a UDP receptor upregulated during brain injury and associated with chemotaxis and phagocytosis [85].

The ADP/ATP receptor P2Y12 (also known as P2RY12), is now widely regarded as a microglial-specific protein, and defined as a marker of homeostatic microglia (Figure 4D) [86]. Interestingly, this receptor is mainly associated with directed motility, and thus with microglia responding to any abnormal changes in their environment, driving chemotaxis through a mechanism that involves the activation of potassium (K^+^) channels [87]. One of the potential K^+^ channels that has been proposed to interact with the P2Y12 receptor is the potassium two-pore domain channel subfamily K member 13 (KCNK13), also known as two-pore domain halothane-inhibited K^+^ channel 1 (THIK-1). The mechanism as to how this channel interacts with P2Y12 remains unclear. Indeed, while directed motility seems to depend on P2Y12, surveillance motility is dependent on KCNK13, implying their involvement in separate motility mechanisms which do not depend on each other. However, P2Y12 potentiates the activity of KCNK13 by inducing hyperpolarization of the cytoplasmic membrane [87]. More research into the other K^+^ channels expressed in the brain is needed in order to increase our understanding of these mechanisms. This mechanism also challenges the concept of P2Y12 as purely a marker of homeostatic microglia.

In conclusion, there is evidence that both actin reorganization and purinergic signaling are involved in the two forms of motility described above (surveillance and directed motility), but we currently lack detailed knowledge of the molecular events and proteins that regulate each form of motility. Deeper insight into the expression of motility-related proteins is necessary for the characterization of microglial behavior in health and disease.

In this review, we are focusing on the morphological and motility changes of microglia during ageing and in the context of Alzheimer’s disease.

## 4. Microglial Morphology and Motility in Ageing

Microglia undergo changes in their morphology and dynamic behavior as part of the ageing process. For instance, a study involving focal injury to the cerebral cortex in aged mice, described increased densities of microglia with hypertrophied cytoplasm, decreased ramification of branches and thickening of processes compared to young animals [88]. Similar findings were reported in another study assessing retinal microglia in aged mice [89], with increased densities of microglia in the inner and outer pigment layers of the retina. In addition, the cells had less branching as well as a reduction in total process length. These findings are consistent with the observation that there was hypertrophy of the microglial cell body in the hippocampus of both aged gerbils and dogs [90,91].

Morphological changes have also been reported in human microglia. Microglial cells can display what has been referred to as “dystrophic” morphology, which is defined by the presence of discontinuous or punctuated processes. The number of microglia with this dystrophic morphology is increased in the aged brain [92]. Of note, dystrophic microglia have been described only in the context of Iba1 immunolabeling, and have not been observed with other microglial markers [93]. Indeed, it was originally thought that the discontinuity of the Iba1 staining implied a cytoplasmic fragmentation (cytorrhexis), in the sense that processes were “breaking down”, separating from the cell body. However, recent evidence has confirmed that processes that appear fragmented with Iba1 staining are actually complete when observed by double-staining for Iba1 and major histocompatibility class (MHC)-II or CD68. Imaging using electron microscopy confirmed this observation, revealing that processes that appear fragmented on light microscopy still remain connected when observed at higher resolution [93]. This suggests that the dystrophic morphology does not represent a rupture of the cell processes but rather a re-distribution of Iba1 localization which might be indicative of a cytoskeletal impairment [93] and, consequently, dysfunctional microglial motility. A further study using digital 3D-reconstruction of Iba1-positive microglial cells found age-associated alterations in microglial morphological features, such as a significant reduction in microglial process length, branching, and grey matter parenchyma coverage in older humans (~82 years old). Interestingly, in contrast to most other animals studied, no difference in microglial numbers or density was found [94].

Regarding microglial motility, it was observed using in vivo time-lapse microscopy, that microglia from aged mice have reduced process motility compared to young mice, in both physiological and pathological (laser-induced injury) conditions [89].

Taken together, these studies, performed in different brain regions, support the concept that the effects of ageing on microglial morphology and motility-related responses are widespread rather than region-specific.

## 5. Age-Related Alterations in Microglial Motility-Related Proteins and Pathways

As microglial motility is the result of the interaction between several proteins, including actin-related proteins and membrane receptors, ultimately, the age-related alterations in microglial morphology and motility discussed above are a result of changes that occur at the molecular level.

Changes in actin levels have not been evaluated in microglia during ageing; however, observations from other cells of the immune system have revealed that F-actin levels are higher in lymphocytes from aged people, and stimulus-induced actin polymerization is lower in neutrophils from this age group, suggesting ageing has an effect on actin dynamics in immune system cells [95,96].

A study assessing the hippocampal transcriptome in rats found decreased CORO1A expression in aged individuals [97]. Changes in the expression or activation pattern of actin-associated proteins can induce cytoskeletal deterioration. A study in aged rhesus monkeys (*Macaca mulatta*) reported increased expression of calpain 1 (CAPN1) in microglia. CAPN1 is a Ca^2+^-activated protease, which participates in the cleavage and degradation of several cytoskeletal proteins, such as TLN1, ACTN1, FLNC, and spectrin. Overexpression of CAPN1 might contribute to cytoskeletal disturbances, thereby impacting on cell motility [98,99].

Age-related differences in the actin assembly and disassembly machinery have been shown with, for example, modification in the expression and distribution of the actin cross-linking protein Iba1 in aged animals (see above section) [88,90,91]. In mice, there is an age-dependent inactivation of SSH1, the CFL1 activator. SSH1 inactivation subsequently causes a decrease in CFL1 activity and an increase in phosphorylated CFL1 [100]. In rat brain, CFL1-actin rod-shaped aggregates were identified in old but not in young animals. Interestingly, in primary cultured rat neurons transfected with human CFL1, these rods were found to block intracellular transport, with abnormal distribution of early endosomes and mitochondria observed in their proximity. Furthermore, rods induced synaptic loss, as observed by the reduced expression of pre- and post-synaptic markers (SV2 and PSD-95, respectively), and confirmed by electrophysiological records.

Studies evaluating age-related transcriptional changes in human microglia have revealed decreased expression of several actin-interacting genes, including *CORO1A*, *TLN1*, *PFN1* (Profilin 1), *EVL* (Enah/Vasp-Like), *ARPC1A*, and *ARPC1B* (actin related protein 2/3 complex subunit 1 and 2), *CAP1* (cyclase associated actin cytoskeleton regulatory protein 1) and *CTNNA2* (catenin alpha 2) [101]. The functional significance of these findings remains to be explored.

Studies that assessed differences in the expression or activity of chemotactic receptors in microglia during ageing in the absence of pathology or in other conditions are largely lacking. The most studied component is P2Y12, which has been suggested to be increased at the mRNA level in aged murine microglia [89]. An age-related differing response to ATP, the P2Y12 agonist, was also noted in mice. Indeed, microglia from younger mice showed rapid extension of existing processes and formation of new processes towards the stimulus, whereas microglia from older mice showed retraction of processes from the stimulus. On washout of the ATP stimulus, both microglia independently of age extended their processes; however this was more prominent in microglia from aged mice [89]. In humans, P2Y12 has been reported expressed by microglia throughout life, in physiological conditions, with specific age-related changes not studied yet [102].

## 6. Microglial Morphology and Motility in Alzheimer’s Disease

Overall, the microglial changes associated with ageing have been noted to be more pronounced in presence of AD pathology, as evidenced in experimental models. In CRND8 mice, a mouse model of amyloidopathy expressing the human Swedish (KM670/671NL) and Indiana (V717F) APP mutations, microglia in the proximity of Aβ plaques were less ramified, with a decreased cell area and fewer junctions compared to CRND8 microglia distant from Aβ plaques and those in wild-type mice [103]. This implies that the changes in morphology may be driven by the presence of the plaques. Similar findings were observed in the human AD brain, with a reduction in total cell volume, total number of branches, and branch length. Other features associated with AD included decreased microglial arborization in the inferior temporal cortex and cingulate cortex suggested to be due to a reduction in the length of branches [94]. In keeping with the known disease progression through the cerebral cortex [104,105], some of the morphological microglial changes, such as the dystrophic morphology, were more pronounced in the inferior temporal cortex compared to the cingulate cortex [94]. This is consistent with a previous study which described the microglial dystrophic morphology to be mainly associated with tau pathology, (dystrophic neurites and neurofibrillary tangles), pointing out a link between microglial impairment and tau accumulation in AD [106].

In a further study in post mortem human brain, to explore the relationship between changes in microglial morphology and disease progression in AD, comparison of microglial morphology and number was performed in different brain areas including the inferior temporal cortex, superior frontal cortex, and primary visual cortex [107]. These anatomical regions were selected based on the kinetics of the pathogenesis of the disease, with the inferior temporal cortex affected early in AD, followed by the superior frontal cortex and then the primary visual cortex. Therefore, the authors considered the inferior temporal cortex analogous to late stage of AD and the primary visual cortex similar to the condition in early AD. Using Iba1 staining, a decreased number of ramified microglia was reported in the temporal cortex, as well as an overall lower number of microglial cells. As in other studies, there was decreased arborization area in temporal and frontal cortex. Interestingly, Iba1-positive microglia showing an activated morphology were observed only in the primary visual cortex suggesting that microglia may be activated mainly during the early stages of AD. Clusters of microglia were detected in association with tau and Aβ in the frontal cortex. In the visual cortex, microglia were associated with tau pathology, with an inverse relationship between ramified microglia and arborized area. Again, this finding supported an evolving role of microglia in disease progression, given the aforementioned assumption that the primary visual cortex would mimic early AD compared to the inferior temporal cortex representing the late stage of the disease [107].

In a two-photon imaging study using the APPPS1 mouse model of AD, a model of amyloidopathy expressing human APP with the Swedish mutation and human mutated presenilin (PS)1 (L166P mutation), microglial motility was impaired in the presence of Aβ plaques when a focal lesion was induced by a laser, with reduced extension of processes towards the lesion. In situ experiments using acute brain slices demonstrated that the speed of microglial process motility was reduced in the APPPS1 compared to control mice after a laser insult [108]. This demonstrated that microglial motility was partly affected by Aβ plaques, impairing the microglial ability to respond in the context of an additional insult.

## 7. Pathological Changes in Actin-Related Proteins in Alzheimer’s Disease

Actin-interacting proteins in microglia undergo changes during ageing and thus potentially may play a role in AD progression. For instance, in a mutant human tauopathy *Drosophila melanogaster* model (R406W), the following features were observed: Abnormal accumulation and aggregation of F-actin, CFL1--actin rods co-localized with phosphorylated tau, and in vitro incubation of F-actin with bovine tau generated bundling of F-actin [109]. These findings are interesting in the light of tau being also a cytoskeletal protein, suggesting that the impairment of microtubule function due to tau hyperphosphorylation may additionally promote dysregulation of the actin dynamics. A reduction in levels of activated CFL1 was observed in APPPS1 mice in an age-dependent manner, exceeding that seen in ageing in wild-type mice. This was associated with increased phosphorylated CFL1 (pCFL1) and a decrease in SSH1, suggesting inactivation of CFL1 may in part be due to a loss of SSH1 activity [99].

Regarding other actin-interacting proteins described in this review, there are few studies exploring Iba1 in the context of its role as an actin cross-linker and microglial motility. One human study found Iba1 expression to be positively associated with the absence of dementia and a good Mini Mental State Examination (MMSE) score. Of note, this study found Iba1 expression correlated with Aβ plaque load, in both control and AD groups [11], consistent with a response to accumulation of Aβ during ageing [110]. In addition, the microglial profile was associated with the apolipoprotein E (APOE) polymorphism, the main genetic risk factor for sporadic AD. Possession of the APOEε2 allele, known to lower the risk of AD, was associated with higher expression of Iba1; whereas possession of the APOEε4, known to increase AD risk, was related to lower expression of Iba1 [11]. Therefore, it could be suggested that APOE as a risk factor in AD acts partly via the impact of apoE phenotype on microglial motility, an important function of physiological microglia, thus linking two main pathways associated to AD.

As described in the APPPS1 mice, CFL1 downregulation, increased pCFL1, and decreased SSH1 have also been seen in human AD [100]. However in contradiction to these findings, another human study found reduced levels of pCFL1 in AD relative to controls [111]. Both studies have investigated CFL1 expression in very few human samples (4 to 7 AD vs. 5 to 4 controls, respectively) and thus their findings might reflect the variability of human brain, emphasizing the requirement for more powerfully designed human studies. One study in human AD has described the presence of CFL1 rods in relation to tau pathology in the hippocampus and inferior temporal cortex independently of age. Unlike the findings in *Drosophila*, phosphorylated-tau and CFL1 rods had less than 5% co-localization [112], possibly due to a time-frame difference in the formation of the two events, with CFL1 rod formation being earlier than tau pathology. The differences between these results and those of the *Drosophila* study also highlight the difficulty of replicating the kinetics of a chronic human disease in animals with a shorter lifespan.

## 8. Changes in Microglial Membrane Chemotactic Receptors in Alzheimer’s disease

Alterations in the proteins that microglia use to perceive and react to their environment seem to also occur in AD. Purinergic signaling has been particularly studied in this context. For instance, upregulation of P2X7 was observed in proximity to Aβ plaques in an AD mouse model, the Tg2576 mice [113], a transgenic mouse model expressing human APP with the Swedish mutation. Similarly, P2X7 expression was reported to be increased in human AD brains, mainly in microglia in proximity to Aβ plaques, with its expression correlated with Aβ load [114]. The role of P2X7 in microglial activation was demonstrated by showing that injection of Aβ into the brain increased intracellular calcium, ATP release, and IL1β secretion in wild-type mice but not P2X7-KO mice. Furthermore, injection of Brilliant Blue G, a P2X7 receptor antagonist, attenuated gliosis, reduced P2X7 receptor expression and decreased blood-brain-barrier leakiness in rats which had intra-hippocampal injection of Aβ_42_, suggesting a neuroprotective role in P2X7 antagonism [115].

The role of the P2X7 receptor in AD has been investigated recently in a study using both experimental models and nine human cases, five of them with confirmed AD. In an APPPS1-P2X7-KO mouse model, decreased Aβ plaques and soluble Aβ relative to APPPS1 mice were detected, and this was associated with improved memory behavior due to rescue of synaptic dysfunction. In humans, P2X7 expression was confirmed in microglia but also in astrocytes. This study further looked at changes in microglial function in relation to P2X7 and identified a role for this purinergic receptor in Aβ -mediated release of chemokines, particularly CCL3, while no effect was observed in IL1β secretion [116]. These findings were in contradiction to prior studies [77,117]. One explanation for this was the design of the previous studies which may have promoted an increase in IL1β cytokine due to external microglial priming following LPS or Aβ injection. Another hypothesis to explain the conflicting results was that P2X7 has differing roles in acute and chronic exposure to Aβ with (i) in acute settings, activation of microglia and (ii) in chronic settings, chemokine secretion. Considering this, the role for P2X7 in human AD remains unclear.

Regarding the metabotropic receptors P2Y, some studies have explored their behavior in AD mouse models. In APPPS1 mice, P2Y1 expression has been associated with astrocyte hyperactivity and spatial learning and memory [118]. In the transgenic AD model CRND8, heterozygous P2Y2-KO mice had decreased microglial CD11b expression in the cerebral cortex and hippocampus and increased soluble Aβ and plaques, compared to age-matched TgCRND8 mice. These P2Y2-KO mice also showed neurological deficits at 10 weeks, such as an abnormal limb-clasping effect and impaired gait, which are not observed in non-KO CRND8 mice, and their life expectancy was reduced to 12 weeks. Interestingly, homozygote mice survived less than 5 weeks. These findings suggest an important role for P2Y2 in neuroprotective mechanisms partly via microglial activation and recruitment [119].

In human AD brains, P2Y2 expression was reduced in parietal cortex, but preserved in the occipital cortex. The changes noted were associated with neuritic plaques and neurofibrillary tangles, and thus with amyloid and tau pathology, but there was no association between P2Y2 and MMSE scores or disease duration. P2Y4 and P2Y6 were also explored in this study with no changes in their expression in AD [120]. In humans, P2Y12 was not detected in microglia clustering around Aβ plaques, and was only found some distance from the plaques and in Iba1-positive cells. Two hypotheses were suggested by the authors with (i) P2Y12 is downregulated in activated/pathogenic microglia as the cells lose their homeostatic phenotype [121,122], or (ii) due to blood-brain-barrier disruption, microglia were replaced by peripheral infiltrating myeloid cells [102]. However, this study included two multiple sclerosis cases and three AD cases, which are diseases with very different pathophysiological pathways. Indeed, recruitment of peripheral macrophages in AD remains to be confirmed in humans.

## 9. Conclusions

Microglia are involved in multiple functions in the brain, both homeostatic and in response to ageing and disease. Changes in motility are associated with the functional state of the cell. Motility is governed by a number of actin-interacting proteins and membrane receptors. In ageing, changes have been identified in several of these proteins, likely reflecting changes in microglial motility. In AD these changes are exacerbated (Table 1), which may contribute to the development and progression of the disease.

Several changes in actin-interacting proteins have been reported to be associated predominantly with tau in experimental models and human AD; whereas changes in purinergic receptors appear to be more commonly associated with Aβ. This is consistent with the nature of Aβ, which is an extracellular peptide more prone to stimulate the “sensing” machinery of the microglial cell. The relationship with tau remains to be clarified. However, it is unclear whether this is a true dichotomy, due to the relative paucity of studies on microglial motility proteins in AD. This review also highlights gaps in the current understanding of the role of certain actin-interacting proteins and purinergic receptors in ageing and AD (Table 1) and the need for further research.

## Figures and Tables

**Figure 1 cells-08-00639-f001:**
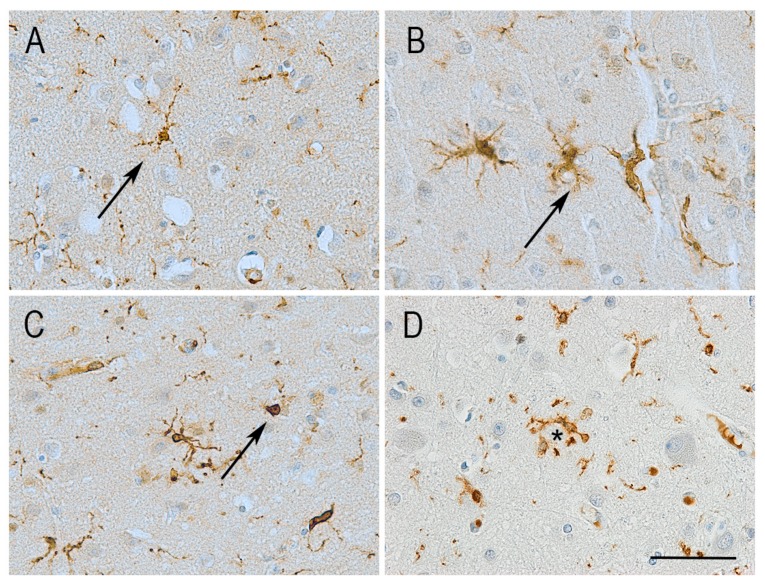
Illustration of the different morphologies adopted by microglia in the human brain independently of age or disease. Immunolabelling for the microglial protein Iba1 shows diverse morphologies including varying number of processes and cell body shape. (**A**) Ramified microglia with small round cell body and several long branching processes. (**B**) Reactive microglia with increased cell body size and reduced length of processes. (**C**) Amoeboid microglia with enlarged cell body and no processes. Images A–C taken from a control aged brain. (**D**) Microglia clustering around Aβ plaques (*) is a feature observed only in the presence of Alzheimer-type pathology. Haematoxylin counterstaining. Scale bar = 50 μm.

**Figure 2 cells-08-00639-f002:**
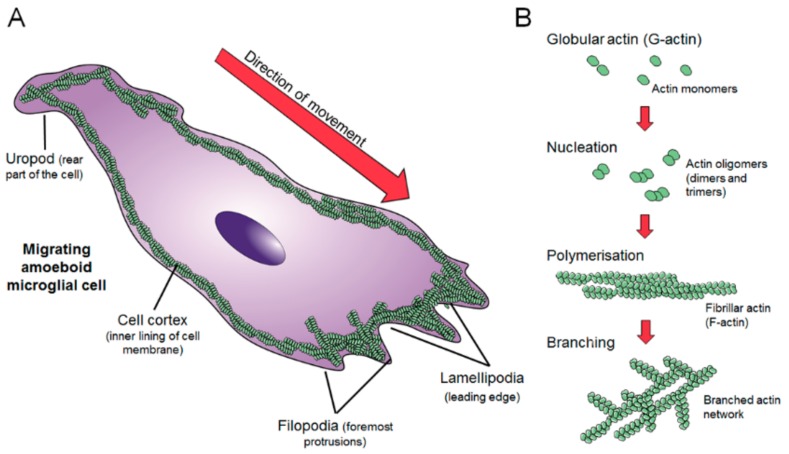
(**A**) Depiction of the different actin structures present in microglia: The cell cortex (covering all the inner surface of the cell), filopodai and lamellipodia (at the leading edge), and the uropod (at the rear of the cell). (**B**) Mechanism of formation of the actin network includes globular actin nucleates in the form of oligomers which further polymerize into left-handed two-chained helical filaments. Filaments additionally recruit globular actin to form branches, which extend from the mother filament at a characteristic 70° angle enabling filaments to easily connect with each other forming an intricate and highly plastic network.

**Figure 3 cells-08-00639-f003:**
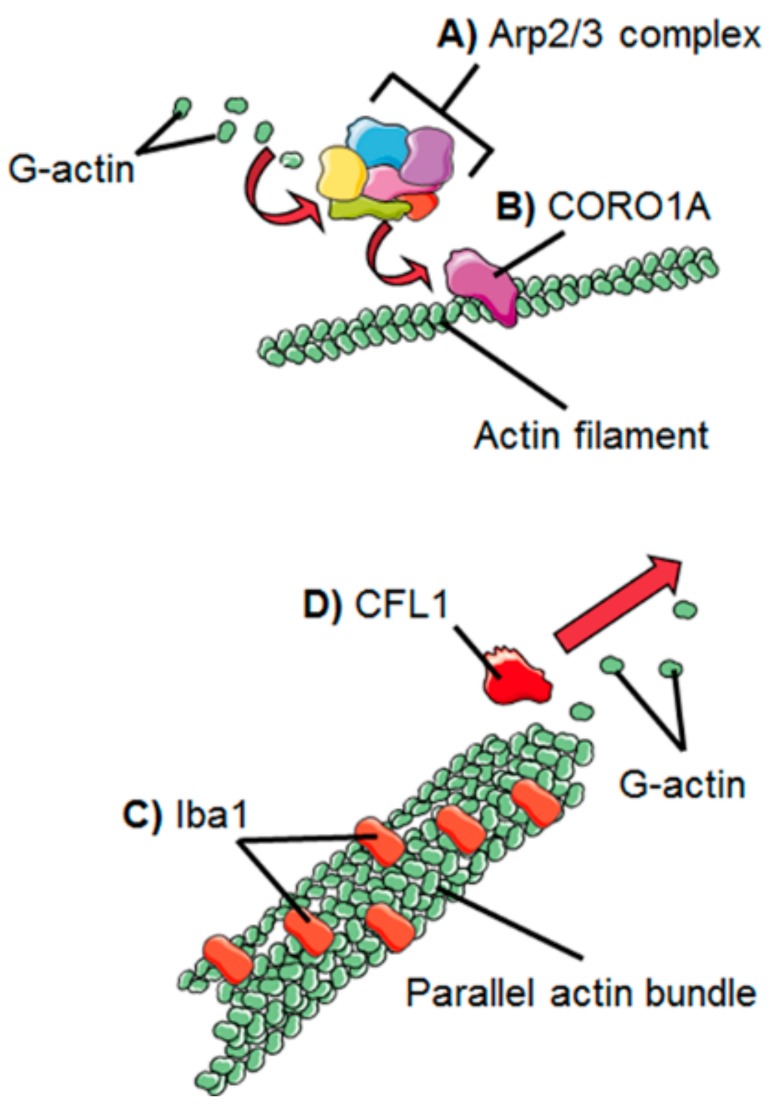
Representation of the mechanisms of action of microglial actin-interacting proteins. (**A**) The process of branching is regulated by the Arp2/3 complex, (**B**) which is recruited by CORO1A to an existing actin filament. Arp2/3 engages with G-actin to form a filament branch. (**C**) Iba1 promotes the formation of parallel actin bundles, scaffold-like structures that give shape to lamellipodia and filopodia. (**D**) CFL1 depolymerizes filaments to make G-actin available to form new actin structures.

**Figure 4 cells-08-00639-f004:**
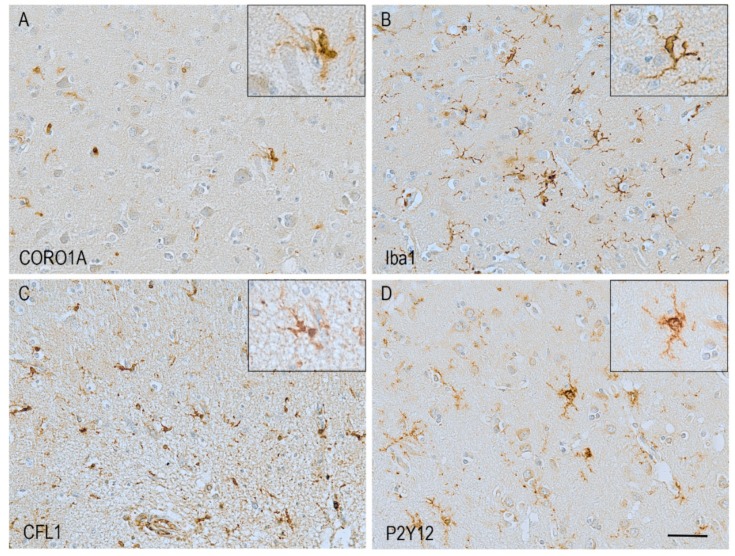
Examples of microglia identified using different motility-related microglial proteins. Haematoxylin counterstaining. **A**—CORO1A; **B**—Iba1; **C**—CFL1; **D**—P2Y12. Scale bar = 50 μm.

**Figure 5 cells-08-00639-f005:**
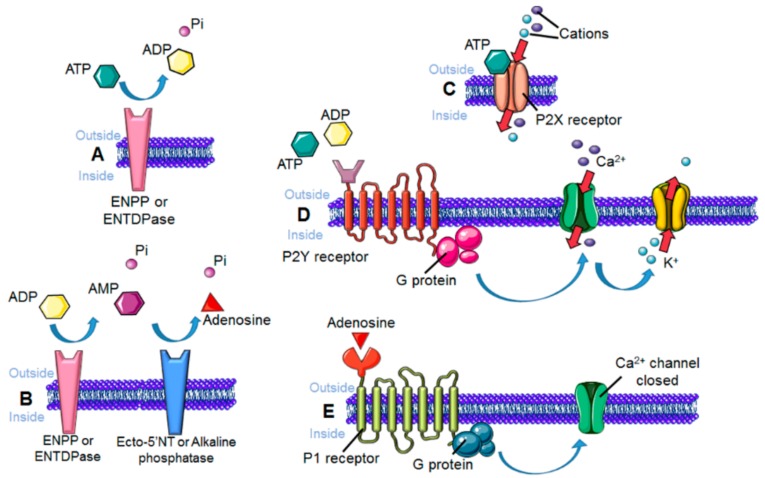
Mechanism of activation of purinergic receptors. (**A**) Extracellular ATP is hydrolyzed to ADP by the action of ectonucleotide pyrophophatase/phosphodiesterases (ENPPs) or ectonucleoside triphosphate dyphosphohydrolases (ENTDPases). (**B**) ADP is subsequently hydrolyzed to AMP, also by ENPPs and ENTDPases. AMP is converted to adenosine by ecto-5′-nucleotidases (Ecto-5′-NTs) or alkaline phosphatases. (**C**) P2X cation-permeable ionotropic receptors are activated by nucleosides triphosphate. (**D**) G protein-coupled P2Y receptors regulate voltage-gated Ca^2+^ and K^+^ channels. (**E**) Adenosine-mediated P1 receptor activation results in blockade of Ca^2+^ channels.

**Table 1 cells-08-00639-t001:** Summary of the most relevant proteins involved in microglial motility: Physiological function and changes in ageing and Alzheimer’s disease highlighting the gaps in our current knowledge.

Protein	Physiological Role	Changes in Ageing	Changes in AD and Animal Models of AD
Cytoskeletal Proteins			
Actin	Main constituent of cytoskeletal microfilaments.	*Animals:* unknown. *Humans:* Increased F-actin level; decreased stimulus-induced actin polymerization.	*Animals:* Presence of CFL1-actin rods. *Humans*: unknown
Arp2/3 complex	Controls branching of actin filaments.	*Animals:* unknown. *Humans:* Decreased expression of subunits ARPC1A and ARPC1B.	unknown
CORO1A	Recruits Arp2/3 to the ends of actin filaments to initiate branching.	*Animals:* Decreased expression. *Humans:* Decreased expression.	unknown
CAPN1	Cleaves and degrades cytoskeletal proteins, such as TLN1, ACTN1, FLNC and spectrin.	*Animals:* Increased expression in aged monkeys and mice. *Humans:* unknown.	unknown
TLN1	Mediates integrin-cytoskeleton bonds, important for cell adhesion.	*Animals:* unknown. *Humans:* Decreased expression.	unknown
Iba1	Involved in actin bundling and membrane ruffling.	*Animals:* Increased in aged gerbils, dogs and mice. *Humans:* Positively correlated to MMSE score.	*Animals:* Increased in APPPS1 mouse model. *Humans:* Positively correlated to MMSE score.
NM II	Essential for the contractile properties of the cytoskeleton.	unknown	unknown
CFL1	Depolymerizes and severs actin filaments.	*Animals:* Increased load of pCFL1, presence of CFL1-actin rods. *Humans:* unknown.	*Animals:* Increased load of pCFL1, presence of CFL1-actin rods. *Humans:* Increased load of pCFL1, presence of CFL1-actin rods.
SSH1	Dephosphorylates CFL-1, inducing activation.	*Animals:* Decreased activity in aged mice. *Humans:* unknown.	*Animals:* Decreased activity in aged mice. *Human:* unknown.
**Chemotactic Receptors**			
CX3CR1	Binds to fractalkine, a neuron-secreted chemokine. Involved in baseline and directed motility and used as a marker specific of microglia.	unknown	unknown
P1 family	A2A involved in process retraction, A3 in process extension.	unknown	unknown
P2X4	ATP/ADP receptor involved in chemotaxis.	unknown	unknown
P2X7	ATP receptor; regulates IL1 secretion.	*Animals:* Increased in aged mice. *Humans:* unknown.	*Animals:* Upregulated in APPPS1 mice and associated with Aβ plaque load. Increased expression after Aβ_42_ intrahippocampal injections in WT mice. Decreased Aβ plaques and soluble Aβ and improved memory in APPPS1-P2X7 KO mice. *Humans:* Increased expression and noted in proximity to Aβ plaques. Upregulated and correlated with Aβ plaque load.
P2Y1	ADP receptor involved in chemotaxis; indirectly influences CFL1 activity.	unknown	*Animals:* In APPPS1 mice, its blockade improved spatial learning and memory. *Humans:* unknown.
P2Y2	UTP receptor, regulates levels of intracellular calcium, involved in chemotaxis.	unknown	*Animals*: CRND8 P2Y2-KO mice had increased soluble Aβ and plaques, and shortened lifespan. *Humans:* Reduced expression.
P2Y4	ATP receptor involved in pinocytosis.	unknown	*Animals:* unknown. *Humans:* No changes found.
P2Y6	UDP receptor, associated with chemotaxis and phagocytosis.	unknown	*Animals:* unknown. *Humans:* No changes observed.
P2Y12	ATP/ADP receptor associated with directed motility.	*Animals:* Increased expression in aged mice. *Humans:* unknown.	*Animals:* unknown. *Humans:* Downregulated in microglia clustered around Aβ plaques.

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
