# Peer review of "Molecular Mechanisms of Microglial Motility: Changes in Ageing and Alzheimer’s Disease"

_cells, 2019, doi:10.3390/cells8060639_

Round 1

Reviewer 1 Report

In the adult brain, microglia perform a constant surveillance of the brain parenchyma, where they constitute the first line of defence against pathogens or injury. Besides their role in immune response, evidence from studies in mice suggests

that microglia have a role in synaptic plasticity, pruning dendritic spines and presynaptic axon terminals. 

In this review, the authors focus on the morphological and motility changes of microglia during ageing and in the context of Alzheimer’s disease. This is an excellent written review. Can be published in its currant form.

Author Response

In this review, the authors focus on the morphological and motility changes of microglia during ageing and in the context of Alzheimer’s disease. This is an excellent written review. Can be published in its currant form.

We thank the reviewer for the supportive comments.

Reviewer 2 Report

This is a well-written review of cytoskeletal proteins in microglia, and also of changes of microglia in aging. The only quibble is that the two do not clearly line up. This is exemplified in the table where there are a large number of "unknowns". This is not necessarily a major fault: we genuinely do not know what changes there will be in microglia of eugeric aged humans let alone in AD.

Author Response

This is a well-written review of cytoskeletal proteins in microglia, and also of changes of microglia in aging. The only quibble is that the two do not clearly line up. This is exemplified in the table where there are a large number of "unknowns". This is not necessarily a major fault: we genuinely do not know what changes there will be in microglia of eugeric aged humans let alone in AD.

Response

We recognised that this is a major gap in our current knowledge and we have taken the opportunity to emphasize this point in the title of the table 1 by modifying the Table 1 description as follows:

Table 1. Summary of the most relevant proteins involved in microglial motility: Physiological function and changes in ageing and Alzheimer's disease highlighting the gaps in our current knowledge.

Reviewer 3 Report

It is a well organized and clear review. Also, the figures and the table help to build the story.

Author Response

It is a well organized and clear review. Also, the figures and the table help to build the story.

We thank the reviewer for the supportive comments.

Reviewer 4 Report

This review article by Franco-Bocanegra et al nicely describes the molecular mechanisms of microglial motility in physiological conditions. Interestingly, the authors also summarize the differential alteration of microglial morphology/mobility during aging and in Alzheimer’s disease. I have some suggestions mentioned below.

1.      Since the selling point of this review is microglial mobility alteration during aging and Alzheimer’s disease, the authors should expand the aging and Alzheimer’s disease sections, including the differential alteration/response of microglial morphology/mobility under pathological conditions in aged microglia vs young microglia and AD microglia vs normal microglia. Figures illustrating the differential alteration of microglia in aging and AD will be appreciated.

2.      Microglia have been found to be brain region-dependent diversity and selective regional sensitivities to aging and diseases, it would be great if the authors can also summarize microglial mobility alteration in different brain regions during aging and AD in the text and a figure.

3.      It would be nice if the authors can provide binary images for microglial morphology in Figure 1 along with description of the features for those microglia in the figure legend.

4.      In figure 2, the authors are suggested to describe action model of microglial movement in detail in the figure legend.

Round 2

Reviewer 4 Report

The authors addressed my main concerns, the revised version of the manuscript appears to be good.